# Concerning the Role of *σ*-Hole in Non-Covalent Interactions: Insights from the Study of the Complexes of ArBeO with Simple Ligands

**DOI:** 10.3390/molecules26154477

**Published:** 2021-07-24

**Authors:** Stefano Borocci, Felice Grandinetti, Nico Sanna

**Affiliations:** 1Dipartimento per la Innovazione nei Sistemi Biologici, Agroalimentari e Forestali (DIBAF), Università della Tuscia, L.go dell’Università, s.n.c., 01100 Viterbo, Italy; borocci@unitus.it (S.B.); n.sanna@unitus.it (N.S.); 2Istituto per i Sistemi Biologici del CNR, Via Salaria, Km 29.500, 00015 Monterotondo, Italy; 3Istituto per la Scienza e Tecnologia dei Plasmi del CNR (ISTP), Via Amendola 122/D, 70126 Bari, Italy

**Keywords:** *σ*-hole, bonding analysis, noble-gas chemistry, non-covalent complexes, SAPT analysis

## Abstract

The structure, stability, and bonding character of some exemplary *L*Ar and *L*-ArBeO (*L* = He, Ne, Ar, N_2_, CO, F_2_, Cl_2_, ClF, HF, HCl, NH_3_) were investigated by MP2 and coupled-cluster calculations, and by symmetry-adapted perturbation theory. The nature of the stabilizing interactions was also assayed by the method recently proposed by the authors to classify the chemical bonds in noble-gas compounds. The comparative analysis of the *L*Ar and *L*-ArBeO unraveled geometric and bonding effects peculiarly related to the *σ*-hole at the Ar atom of ArBeO, including the major stabilizing/destabilizing role of the electrostatic interactionensuing from the negative/positive molecular electrostatic potential of *L* at the contact zone with ArBeO. The role of the inductive and dispersive components was also assayed, making it possible to discern the factors governing the transition from the (mainly) dispersive domain of the *L*Ar, to the *σ*-hole domain of the *L*-ArBeO. Our conclusions could be valid for various types of non-covalent interactions, especially those involving *σ*-holes of respectable strength such as those occurring in ArBeO.

## 1. Introduction

Non-covalent interactions (NCIs) play a major role in natural and synthetic processes [1,2,3]. Over the years, the traditional interest for the hydrogen bond (HB) [4,5] has been progressively extended to other NCIs, and greatest attention is, in particular, currently paid to *σ*-hole [6,7,8,9] interactions. The tetrel, pnictogen, chalcogen, and halogen bonds [10,11] (the latter actually rivaling in interest [12,13] that of the HB) are just elder sisters of a large family that embraces most groups of the periodic table [14,15].

*σ*-holes are regions of charge depletion centered on the outer extension of covalent bonds. They are typically associated with maximum points (*V*_S,max_) of the molecular electrostatic potential (MEP) [16] (customarily taken at or at around the 0.0010 *ea*_0_^−3^ isodensity surface [17]), and the value of the MEP at *V*_S,max_ is generally (but not always) positive. Thus, *σ*-holes typically behave as electrophilic centers, able to promote by contact with a ligand a complex blend of electrostatic, polarization (including induction and charge transfer), and dispersion contributions. The detailed assay of these terms, and of their dependence on the type and the strength of the interaction, are, indeed, of major interest in the study of NCIs. In a recent study [18], we sought to acquireinsights into *σ*-hole effects by exploring the changes occurring when pure (or nearly pure) dispersive contacts wereconverted into *σ*-hole interactions. The investigated systems were inspired, in particular, by already-established evidence concerning the complexes of the noble-gas atoms (Ng) with beryllium Lewis acids, particularly BeO [19]. It is, thus, well known that the NgBeO feature remarkably high binding energies, mainly arisingfrom the appreciable polarization of Ng by BeO, and the ensuing appreciable inductive stabilization. We thus surmisedthat such a major electronic effect could induce a *σ*-hole on Ng, andexplored the MEP of the NgBeO. The calculations actually confirmed a *σ*-hole already appreciable in ArBeO, and only slightly more pronounced in KrBeO and XeBeO. We thus focused on the simplest *L*-ArBeO (*L* = He, Ne, Ar, HF), and compared their bonding situation with that of their “dispersive” cousins, *L*Ar [18]. To this end, we employed the symmetry-adapted perturbation theory (SAPT) [20,21], and the method that we recently proposed [22,23,24] to analyze the bonding situation of Ng compounds. Compared with the *L*Ar, the *L*-ArBeO were unraveled as structurally more compact, and this mirrored increased values of the SAPT interaction energies (with respect to the *L* + Ar and *L* + ArBeO dissociation limits). The two most affected binding components were, in particular, the electrostatic and the inductive, theirincreases seemingly being related to the electrostatic potential of *L*, and to its polarizability [18]. Further evidence in this regard came from the failed location of a bound FH-ArBeO, that we ascribed [18], likewise to other “counterintuitive” NCIs [25,26],to the unfavorable electrostatic contact between the positive H atom of HF and the positive Ar atom of ArBeO. To reinforce the interpretation, we decided to explore other exemplary *L*Ar and *L*-ArBeO, including *L* = N_2_, CO, F_2_, Cl_2_, ClF, HCl, and NH_3_. The MEP of these ligands ranges, in fact, between definitely-negative to definitely-positive values, and this should magnify any effect arising from the contact with the *σ*-hole of ArBeO. The obtained results, discussed in the present article, actually confirmed this expectation, and furnished insights that could be useful in discussing other types of*σ*-holecomplexes.

The paper is organized as follows. Section 2 and Section 3 provide, respectively, a brief account of the methods employed to perform the bonding analysis, and the most relevant computational details. Section 4 first presents the investigated *L*Ar and *L*-ArBeO, their predicted data, and an estimate of their accuracy. Then follows the SAPT analysis of the *L*Ar and *L*-ArBeO, the former being discussed first as reference systems to best highlight the *σ*-hole effects occurring in the complexes of *L*-ArBeO. The *L*Ar and *L*-ArBeO are then compared in terms of our proposed descriptors of bonding character. The most relevant conclusions are providedin Section 5.

## 2. Methods of Bonding Analysis 

The bonding analysis was first accomplished by SAPT [20,21]. In this approach, the total Hamiltonian of the dimer is partitioned as *H* = *F* + *V* + *W*, where *F* = *F_A_* + *F_B_* is the sum of the Fock operators for monomers *A* and *B*, *V* is the intermolecular interaction operator, and *W* = *W_A_* + *W_B_* is the sum of the Møller-Plesset fluctuation operators. The latter are defined as *W_X_* = *H_X_* – *F_X_*, where *H_X_* is the total Hamiltonian of monomer X. The $E_{int}^{SAPT}$ is expanded as a perturbative series, and, in particular, we included the following terms:(1)EintSAPT=Eelst10+Eexch10+Eind,r20+Eexch−ind,r20+δEint,rHF+Eelst,r12+Etind22+Etexch−ind22+Edisp20+Eexch−disp20+Eelst,r13+εexch1CCSD+Edisp21+Edisp22
the first (1/2) and the second (0/1/2/3) number superscript in parenthesesindicating the first-/second-order, and the zero^th^-/first-/second-/third-order intramonomer electron correlation correction to *V* and *W*, respectively. The notations in subscript indicate the classical (Coulombic) electrostatic energy (*elst*), the exchange term that results from the antisymmetrization of the wave-function (*exch*), the induction energy (*ind*) (including the charge transfer), and the dispersion energy (*disp*). The “*r*” indicates that a given component has been computed by including the coupled Hartree-Fock (HF) response for the perturbed system. The $\delta E_{int,r}^{HF}$ term collects the contributions to the supermolecular HF energy beyond the second-order of intermolecular operator, the Etind22 is the part of $E_{ind}^{\left(22\right)}$ not included in $E_{ind,r}^{\left(20\right)}$, and $\varepsilon_{exch}^{\left(1\right)}\left(CCSD\right)=E_{exch}^{\left(1\right)}\left(CCSD\right)-E_{exch}^{\left(10\right)}$ is the part of $\varepsilon_{exch}^{\left(1\right)}\left(\infty \right)$ with intra-monomer excitations at the CCSD level of theory.

The terms of Equation (1) were grouped so to express $E_{int}^{SAPT}$ as the sum of the electrostatic ($E_{elst}$), inductive ($E_{ind}$), dispersive ($E_{disp}$), and exchange ($E_{exch}$) components:(2)$$E_{elst}=E_{elst}^{\left(10\right)}+E_{elst,r}^{\left(12\right)}+E_{elst,r}^{\left(13\right)}$$
(3)Eind=Eind,r20+Eexch−ind,r20+Etind22+Etexch−ind22+δEint,rHF
(4)$$E_{disp}=E_{disp}^{\left(20\right)}+E_{disp}^{\left(21\right)}+E_{disp}^{\left(22\right)}+E_{exch-disp}^{\left(20\right)}$$
(5)$$E_{exch}=E_{exch}^{\left(10\right)}+\varepsilon_{exch}^{\left(1\right)}\left(CCSD\right)$$

Our method of bonding analysis [22,23,24] relies on the examination of the plotted shape of the local electron energy density *H*(***r***) [22,27,28], and on the values that this function takes over the volume *Ω*_s_ enclosed by the *s*(***r***) = 0.4 reduced-density gradient (RDG) isosurface [29,30] associated with the bond-critical point (BCP) located for a given Ng-X (X = binding partner) from the topological analysis of the electron density *ρ*(***r***) [31]. Ancillary indices include the size of *Ω*_s_, the total electronic charge enclosed by *Ω*_s_, *N*(*Ω*_s_), the average electron density over *Ω*_s_, *ρ*_s_(ave) = *N*(*Ω*_s_)/*Ω*_s_, and the average, maximum, and minimum values of *H*(***r***) over *Ω*_s_, *H_s_*(ave, max, min). As discussed previously [22], the *H*(***r***) partitions the atomic space in two well-recognizable regions, namely an inner one of negative values, indicated as *H*^–^(***r***), and an outer one of positive values, indicated as *H*^+^(***r***). The boundary of these regions falls at a distance *R*^–^ that is typical of each atom; at this distance, *H*(***r*** = *R*^−^) = 0. Interestingly, when two atoms form a chemical bond, their *H*^−^(***r***) and *H*^+^(***r***) regions combine in modes that signal the nature of the interaction. Particularly for the Ng-X bonds, it is possible to distinguish three types of interactions, indicated as A, B, or C. 

In interactions of type A, the atoms overlap all the contour lines of their *H*^+^(***r***) regions, and part of the contour lines of their inner *H*^−^(***r***) regions, the bond appearing as a continuous region of negative values of *H*(***r***), plunged in a zone of positive values. The interaction is topologically-signed by a (3, +1) critical point of the *H*(***r***) (denoted as the HCP) falling on the bond axis. Exemplary in this regard is the N-N bond of the N_2_ moiety of the N_2_-Ar and the linear N_2_-ArBeO shown in Figure 1. The Ar-Be bond of N_2_-ArBeO is, instead, a showcase of interactions of type B. In these contacts, the *H*^−^(***r***) region of Ng is, again, overlapped with the *H*^−^(***r***) region of the binding partner, but (*i*) no HCP exists on the bond axis, and (*ii*) the Ng-X inter-nuclear region includes a (more or less wide) region of positive *H*(***r***). Finally, the N-Ar bonds of N_2_-Ar and N_2_-ArBeO are illustrative examples of interactions of type C. In contacts like these, the Ng and the binding partner overlap only part of their *H*^+^(***r***) regions, their *H*^−^(***r***) regions remaining, instead, perfectly closed, and separated by a (more or less wide) region of positive *H*(***r***). The bond thus appears as two clearly distinguishable *H*^−^(***r***) regions, separated by a region of positive values of *H*(***r***).

Any Ng-X is assigned as covalent (Cov) if (*i*) it is of type A, and (*ii*) the electron density at the BCP, *ρ(*BCP*)* is at least 0.08 *ea*_0_^−3^. This is the case of the N_2_ moiety of the N_2_-Ar and N_2_-ArBeO shown in Figure 1a,b. Any Ng-X not fulfilling these criteria wasfurther assayed by integrating the *H*(***r***) over *Ω*_s_. If the function was, invariably, positive over the entire volume, the interaction wasassigned as non-covalent (nCov). If the function waspartially or fully negative, the Ng-X wasassigned as partially-covalent (pCov), and distinguished as H^+/−^, H^−/+^, and H^−^, the superscript indicating that, over *Ω*_s_, the *H*(***r***) was ranging from negative to positive, but, on the average, was positive (H^+/−^) or negative (H^−/+^) or, that it is invariably negative (H^−^). Interactions of type C involving a single Ng atom, such as the N_2_-Ar shown in Figure 1a, and all the other presently investigated *L*Ar (*vide infra*) were also assayed in terms of the *degree of polarization* of Ng, DoP(Ng). This index measures, in essence, the deformation of the *H*^−^(***r***) region of Ng arising from the interaction with X [23,32]. The DoPis, in particular, defined by the equation:(6)$$\mathrm{DoP}\left(\mathrm{Ng}\right)=\frac{[R_{\mathrm{Ng}}^{-}\left(\mathrm{Ng}-X\right)-R_{\mathrm{Ng}}^{-}]\times 100}{R_{\mathrm{Ng}}^{-}}$$
where $R_{\mathrm{Ng}}^{-}\left(\mathrm{Ng}-X\right)$ is the radius of the *H*^−^(***r***) region of Ng along the axis formed by Ng and the Ng-X BCP, and $R_{\mathrm{Ng}}^{-}$ is the radius of the *H*^−^(***r***) region of the free atom. The positive/negative sign of the DoP signal Ng atoms polarized toward/opposite to X, and the magnitude of the DoP is related to the extent of the polarization. Illustrative examples of the effective use of this index are providedin Refs. [23,32].

## 3. Computational Details

The employed levels of theory were the Møller-Plesset truncated at the second order (MP2) [33], the coupled cluster with inclusion of single and double substitutions, CCSD, and an estimate of connected triples, CCSD(T) [34]. The calculations were performed with the Dunning’s correlation consistent aug-cc-pVTZ [35] (denoted here as aVTZ) by explicitly correlating the outer valence electrons (frozen-core approximation). The MP2 and CCSD(T) calculations were performed, respectively, with the Gaussian 09 (G09) (Revision D1) [36], and the CFOUR (version 2.1) [37].

The SAPT [20,21] calculations were performed with the SAPT2016 [38], using the G09 for the integrals calculation. The employed basis set, denoted here as aVTZ/mbf, combined the aVTZ with a set of extra 3*s*3*p*2*d*1*f* mid-bond functions [39,40] (three *s* and three *p* functions with exponents 0.9, 0.3, 0.1, two *d* functions with exponents 0.6 and 0.2, and one *f* function with exponent 0.3) placed at the mid-point of any Ar-X distance (X = atom closest to Ar in the linear complexes, or center-of-mass of X_2_ in the T-shaped complexes).

The functions investigated in the bonding analysis werethe *ρ*(***r***) [31], the *H*(***r***) [22,27,28], and the RDG and its related NCI indices [29,30]. The *ρ*(***r***) is defined by the equation [31]:(7)$$\rho \left(r\right)=\underset{i}{\sum }\eta_{i}{\left|\varphi_{i}\left(r\right)\right|}^{2}$$
where $\eta_{i}$ is the occupation number of the natural orbital $\varphi_{i}$, in turn expanded as a linear combination of the basis functions.

The *H*(***r***) is the sum of the kinetic energy density *G*(***r***) and the potential energy density *V*(***r***):(8)$$H\left(r\right)=G\left(r\right)\;+\;V\left(r\right)$$

The presentlyemployed definition [31,41] of the *G*(***r***) is given by the equation:
(9)$$G\left(r\right)=\frac{1}{2}\underset{i=1}{\sum }\eta_{i}{\left|\nabla \varphi_{i}\left(r\right)\right|}^{2}$$ where the sum runs over all the occupied natural orbitals $\varphi_{i}$ of occupation numbers $\eta_{i}$. The potential energy density *V*(***r***) is evaluated [31] from the local form of the virial theorem:(10)$$V\left(r\right)=\frac{1}{4}\nabla^{2}\rho \left(r\right)-2G\left(r\right)$$

The RDG is defined by the equation [29,30]:(11)$$s\left(r\right)=\frac{\left|\nabla \rho \left(r\right)\right|}{2{\left(3\pi^{2}\right)}^{\frac{1}{3}}\times \rho {\left(r\right)}^{\frac{4}{3}}}$$

Low-value *s*(***r***) isosurfaces (typically 0.3–0.6) appear among atoms undergoing any type of interaction, the NCIs emerging, in particular, by considering the spatial regions of low *ρ*(***r***). The low-*s*(***r***)/low-*ρ*(***r***) isosurfaces are, in turn, mapped in terms of the sign (*λ*_2_) × *ρ*(***r***), *λ*_2_ being the second eigenvalue (*λ*_1_<*λ*_2_<*λ*_3_) of the Hessian matrix of *ρ*(***r***). In essence, the sign of *λ*_2_ is used to distinguish between attractive (*λ*_2_< 0) and repulsive (*λ*_2_> 0) interactions, and the value of *ρ*(***r***) is exploited to rank the corresponding strength. In the present study, we also calculated the integral of a given property *P* (particularly the *ρ*(***r***) and the *H*(***r***))over the volume *Ω*_s_ enclosed by the *s*(***r***) = 0.4 isosurface at around the BCP located on any Ar-X bond path, *P*(*Ω*_s_). This integration was accomplished by producing an orthogonal grid of points that enclosed the isosurface and applying the formula:(12)$$P\left(\Omega_{s}\right)=\underset{i\left(\mathrm{RDG}\;<s\right)}{\sum }P\left(r_{i}\right)d_{x}d_{y}d_{z}\;$$
where *P*(***r***_i_) is the value of *P* at the grid point ***r***_i_, and *d_x_*, *d_y_*, and *d_z_* are the grid step sizes in the *x*, *y*, and *z* directions, respectively (*d_x_* = *d_y_* = *d_z_* = 0.025 *a*_0_). The summation is carried out on all grid points ***r***_i_ having RDG < s.

The *ρ*(***r***), the *H*(***r***), and the *s*(***r***) were analyzed with the Multiwfn (version 3.8.dev) [42] using the MP2/aVTZ wave functions stored in the wfx files generated with the G09 or the CCSD(T)/aVTZ wave functions stored in the molden files generated with CFOUR, and subsequently formatted with the Molden2AIM utility [43]. The two-(2D) plots of the *H*(***r***) were also produced with the Multiwfn, and included the standard contour lines belonging to the patterns ±*k* × 10*^n^* (*k* = 1, 2, 4, 8; *n* = −5 ÷ 6), together with the contour lines corresponding to the critical points specifically located from the topological analysis of the *H*(***r***).

## 4. Results and Discussion

### 4.1. The Investigated LAr and L-ArBeO: Predicted Data and Their Accuracy

The investigated *L*Ar includedthe diatomic HeAr, NeAr, and ArAr, the linear and T-shaped (N_2_)Ar, (F_2_)Ar, and (Cl_2_)Ar, the linear isomeric (CO)Ar, (FCl)Ar, (HF)Ar, and (HCl)Ar, and the H_3_N-Ar structure of C_3v_ symmetry. These species were explored at both the MP2/aVTZ and CCSD(T)/aVTZ level of theory, andinvariably characterized as true minima on the potential energy surface (PES). The only exceptions were the OC-Ar and the H_3_N-Ar, both featuring a doublydegenerate imaginary absorption (number of imaginary frequencies NIMAG = 2). As shown in Appendix A, the MP2/aVTZ and CCSD(T)/aVTZ harmonic frequencies of the *L*Ar were, invariably, quite similar, and the MP2/aVTZAr-X distances featured a mean unsigned deviation (MUD) from the CCSD(T)/aVTZ values of only 0.0385 Å.We also ascertained (see Appendix A) the high similarity between the MP2/aVTZ and CCSD(T)/aVTZ predicted data of the simplest He-ArBeO, Ne-ArBeO, Ar-ArBeO, and HF-ArBeO. Based on these findings, all the other *L*-ArBeO (*L* = N_2_, CO, F_2_, Cl_2_, ClF, HCl, NH_3_) were investigated at the predictably accurate MP2/aVTZ. The connectivities and bond distances of all the presently investigated *L*Ar and *L*-ArBeO are thus shown in Figure 2 (their Cartesian coordinates are also availableas Appendix A).Their CCSD/aVTZ*T*_1_ diagnostics (the norm of the vector ***t***_1_ of the single-excitation amplitudes from the CCSD calculation divided by the square root of the number of correlated electrons *N*, $T_{1}=\sqrt{\frac{t_{1}\cdot t_{1}}{N}}$) resulted invariably within the accepted threshold of 0.02 [44], thus confirming the validity of a mono-determinantal description of their wave functions.

The MP2/aVTZ structures of the *L*Ar and *L*-ArBeO were then employed to perform the SAPT calculations (dissociation limits: *L* + Ar and *L* + ArBeO), and the bonding analysis. The obtained results are shown in Table 1 and Table 2. For the *L*Ar, the computed $E_{int}^{SAPT}$ were, in general, well consistent (MUD = 0.02 kcal mol^−1^) with other accurate theoretical estimates already available from the literature [45,46,47,48,49,50,51,52,53,54,55]. In addition, as shown in Appendix A, the results of the bonding analysis of the *L*Ar performed at the MP2/aVTZ and CCSD(T)/aVTZ levels of theory furnished strictly similar results. These findings, overall, support the good accuracy of the data reported in Table 1 and Table 2. Their detailed discussion is providedin the forthcoming paragraphs.

### 4.2. SAPT Analysis of the LAr: The Role of the MEP of L

We first discussthe SAPT data of the *L*Ar(dissociation limit: *L* + Ar). The complexes of Ng with neutral speciesare, in general, perceived as typical van der Waalsmolecules [56], held together by the favorable balance between dispersion and exchange repulsion. The leading interaction components of any *L*Ng may include, however, electrostatic, inductive, and even charge-transfer contributions [57]. Illustrative in this regard are also the presently-investigated *L*Ar, whose decomposed $E_{int}^{SAPT}$(see Table 1) clearlyunraveledan invariably major $E_{disp}$, but also non-negligible contributions of $E_{elst}$ and $E_{ind}$. Especially relevant in the present contextis the dependence of these terms on the MEP of *L*, particularly on the values that this function takes for the various ligands at the *V*_S,max_ or *V*_S,min_ occurring on the outer elongation of the bond axis or perpendicular to it. We could locate these critical points on any probed isodensity surface (namely 0.0005, 0.0010, 0.0015, and 0.0020 *ea*_0_^−3^), and, as shown in Appendix A, the values of the MEP at these points wereonly less sensitive to the employed geometry(experimental, MP2/aVTZ or CCSD(T)/aVTZ), and tothe methodused to calculate the electron density (MP2/aVTZ or CCSD(T)/aVTZ). The forthcoming discussion is based, in particular, on the values quoted in Table 3, computed at the MP2/aVTZ0.0010 *ea*_0_^−3^ isodensity surfaces using the MP2/aVTZ-optimized geometries.

We first note from Figure 2 the strict relationship between the connectivity of any *L*Ar and the position of the *V*_S,max_ or *V*_S,min_ of the involved ligand. In essence, the anisotropy of the electron density of *L* directed the Ar atom toward the stationary point(s) on the PES. Depending on *L*, however, the potential locally-experienced by Ar was quite different, and ranged (see Table 3) from the definitelynegative one at the *V*_S,min_(N) of NH_3_ (−37.25 kcal mol^−1^) to the definitely positive one at the *V*_S,max_(H) of HF (68.78 kcal mol^−1^), passing through the only slightly-positive values of He (1.26 kcal mol^−1^), Ne (1.02 kcal mol^−1^), Ar (1.85 kcal mol^−1^), and at the *V*_S,max_(perp) of F_2_ (0.76 kcal mol^−1^) and Cl_2_ (1.26 kcal mol^−1^). Interestingly, as expected from the results of our recent studies on other numerous Ng complexes [23,32], these different local fields produced a well-recognizable effect, namely the polarization of Ar in modes and extents that strictly mirror the values of the MEP. This phenomenon is effectively caught by the DoP(Ar) (*vide supra*) of the various *L*Ar (see Table 1) that is positively correlated with the MEP at the *V*_S,max_ or *V*_S,min_ of *L* (see Table 3). This is graphically shown in Figure 3.

In essence, when exposed to the MEP of the ligand, the Ar atom polarized toward/opposite to it, the effect increasing by increasing the magnitude of the experienced positive/negative MEP. However, the polarization of Ar was, expectedly, the major physical phenomenon behindthe inductive component of the interaction. As a matter of fact, as shown in Appendix A, for both positive and negative values of the MEP of *L*, the $E_{ind}$ of the various *L*Ar generally increased by increasing the magnitude of the MEP. This positive correlation becomes even more clear by inspecting Figure 4, showing the dependence on the MEP of *L*of the percentage contribution of $E_{ind}$ to the total attractive part of the SAPT interaction, $E_{ind}\left(\%\right)$. On the other hand, neither the absolute $E_{elst}$ nor the percentage contribution $E_{elst}\left(\%\right)$ of the *L*Ar correlated with the MEP of *L*. This was, indeed, not unexpected, as this termsolely arises from non-specific electrostatic interactions between the frozen densities of *L* and of the apolar Ar. One also notes from Table 1 thatthe $E_{elst}\left(\%\right)$ of the various *L*Ar spanned in the small range between ca. 12 and ca. 22%. These nearly constant contributions produced an inverse dependence between $E_{ind}\left(\%\right)$ and $E_{disp}\left(\%\right)$ (Appendix A), a positive correlation between $E_{disp}\left(\%\right)$ and the MEP of *L* (Appendix A), and the positive correlation shown in Figure 5 between the $E_{ind}/E_{disp}$ ratio of the *L*Ar, and the MEP of *L*. The absolute values of $E_{disp}$ are, in any case, totally uncorrelated with the MEP. The dispersion is, in fact, also a form of polarization [67], but different in originfrom the inductive effects exerted by the MEP.

The ligation of BeO to the Ar atom of any *L*Ar produced major structural and stability effects that wereclearly referable to the *σ*-hole occurring at Ar. This is best discussed in the subsequent paragraph.

### 4.3. From the LAr to the L-ArBeO: The Role of the σ-Hole of ArBeO

The only slightly positive electrostatic potential of Ar (1.85 kcal mol^−1^ at the MP2/aVTZ0.0010 *ea*_0_^−3^ isodensity surface) was dramatically affected by the ligation with BeO. As shown in Figure 6a, plotting the MEP of ArBeO in the scale between −20 and 48 kcal mol^−1^, the Ar atom appeared as definitely positive, with a seemingly uniform MEP at around 48 kcal mol^−1^. However, as shown in Figure 6b, using a narrower scale between 42.0 and 50.0 kcal mol^−1^, the MEP of Ar emerged as clearly anisotropic, and featured a *V*_S,max_ of 50.9 kcal mol^−1^ on the outer prolongation of the Ar-Be axis. This maximum signs a *σ*-hole arising from the polarization of Ar by BeO toward the inner region of the Ar-Be bond.

The major ensuing effects are best appreciated by comparing the SAPT data of the *L*-ArBeO(dissociation limit: *L* + ArBeO) with those of the corresponding *L*Ar. We first discuss the changes occurring in the electrostatic component of the interaction. In this regard, based on the data quoted in Table 1, it is possible to recognize three major situations. Thus, when Aris in contact with ligands whose MEP at the *V*_S,max_ or *V*_S,min_ was negative or only slightly-positive (up to less than 2 kcal mol^−1^), the $E_{elst}$ of the *L*-ArBeO was invariably negative, and higher in magnitude than that of the corresponding *L*Ar. The $E_{int}^{SAPT}$ was also definitely more negative than that of the *L*Ar, and the Ar-X distance wasshorterby ca. 0.2−0.4 Å, the contraction arriving up to ca. 0.65 Å for *L* = NH_3_. These systems include the Ng-ArBeO (Ng = He, Ne, Ar), the linear N_2_-ArBeO, OC-ArBeO, CO-ArBeO, ClF-ArBeO, and HF-ArBeO, the T-shaped F_2_-ArBeO and Cl_2_-ArBeO, and the C_3v_ isomer H_3_N-ArBeO.All these species are also characterized as true minima on the MP2/aVTZ PES, the only exception being the T-shaped Cl_2_-ArBeO, featuring a single imaginary frequency (NIMAG = 1) (see Appendix A). According to the electrostatic model of NCIs [67,68], these contacts must be viewed as “intuitive” interactions [25,26] driven from the favorable attraction between MEPs of opposite sign. When Ar was in contact with *V*_S,max_ or *V*_S,min_ featuring positive values of the MEP up to ca. 25 kcal mol^−1^, the $E_{elst}$ of the *L*-ArBeO was lower in magnitude than that of the corresponding *L*Ar, and even positive. The $E_{int}^{SAPT}$ was also less negative than that of the *L*Ar, or even positive. These systems include the T-shaped N_2_-ArBeO, and the linear F_2_-ArBeO and Cl_2_-ArBeO, invariably characterized as higher-order saddle points on the corresponding PES (see Appendix A). All these features are typical of “counterintuitive” complexes [25,26], whose bound characters arise from inductive and dispersive components still sufficient to compensate for the repulsive electrostatic contributionfrom MEPs of the same sign.Finally, the contact of Ar with ligands featuring definitely positive values of the MEP at their *V*_S,max_ was totally repulsive, and the corresponding *L*-ArBeO were unbound on the PES. This happened, in particular, for the FH-ArBeO, FCl-ArBeO, and ClH-ArBeO. The major role of the MEP of *L* in determining these different situations clearly emerges by examining Figure 7, showing the positive correlation between this quantity andthe $E_{elst}$ of the *L*-ArBeO. The *percentage* contributions $E_{elst}\left(\%\right)$, covering the wide range between ca. 9 and ca. 72%, werealso regularly dependent on the MEP of *L* (see Appendix A).

The major role of the electrostatic component in determining the stability of the *L*-ArBeO clearly emerges by examining some exemplary situations. Thus, the higly negative *V_S,Min_*(N) of NH_3_of −37.25kcal mol^−1^produced a *E*_elst_ of the H_3_N-ArBeO as negative as ca. −6.9 kcal mol^−1^ (see Table 1). The $E_{elst}$ of the H_3_N-Ar was, instead, only ca. −0.2 kcal mol^−1^, and the difference of −6.7 kcal mol^−1^ decisively contributed to the huge increase of the $E_{int}^{SAPT}$ when going from H_3_N-Ar (−0.33 kcal mol^−1^) to H_3_N-ArBeO (−5.4 kcal mol^−1^). It is also of interest to compare the N_2_-ArBeO with the isomeric OC-ArBeO and CO-ArBeO. The *V_S,Min_*(N) of N_2_, −8.53 kcal mol^−1^was more negative than the *V_S,Min_*(O) of CO, −4.12, and this produced a $E_{elst}$ of N_2_-ArBeO (−0.98 kcal mol^−1^) that was more negative than that of the CO-ArBeO (−0.90 kcal mol^−1^). This is opposite to the trend across N_2_-Ar and CO-Ar, whose $E_{elst}$ amounts to ca. −0.11 and −0.12 kcal mol^−1^, respectively. Conversely, the *V_S,Min_*(C) of CO of -14.04 kcal mol^−1^produced a $E_{elst}$ of OC-ArBeO (ca. −1.55 kcal mol^−1^) that was more negative than that of N_2_-ArBeO, the difference of −0.57 kcal mol^−1^ decisively determining the $E_{int}^{SAPT}$ of OC-ArBeO as more negative than that of N_2_-ArBeO (−1.49 vs. −1.16 kcal mol^−1^). As a matter of fact, in the absence of this electrostatic stabilization, the $E_{int}^{SAPT}$ of N_2_-Ar and OC-Ar were, essentially, the same (−0.25 and −0.24 kcal mol^−1^, respectively).

As for the inductive component of the interaction, its magnitude invariably increased(becamemore negative) when going from any *L*Ar to the corresponding *L*-ArBeO. At variance with the *L*Ar, however, the values of $E_{ind}$ were not correlated with the MEP of the *L*. This finding was, indeed, not unexpected, as this energy term expectedly mirrors the mutual polarization of *L* and ArBeO. This suggestion is partially supported by the graph plotted in Appendix A, showing the dependence of the $E_{ind}$ of the *L*-ArBeO on the experimental polarizability α of *L* [60,61,62,63,64,65,66,67,68] (see Table 3). For the majority of the complexes, the two quantities were, indeed, positively correlated. There were, however, at least three major deviations (HF-ArBeO, H_3_N-ArBeO, and the linear Cl_2_-ArBeO), whose occurrencesclearly suggestthat various factors conceivably contributed to this energy component. In any case, for any *L*-ArBeO, the *percentage* contribution $E_{ind}\left(\%\right)$, wasonly minor, and spannedover the rather limited range of ca. 9–21% (see Table 1). As shown in Appendix A, this producedan inverse relationship between $E_{elst}\left(\%\right)$ and $E_{disp}\left(\%\right)$, and a positive correlation between the $E_{disp}\left(\%\right)$ of the various *L*-ArBeO, and the MEP of *L*. This also producedthe positive correlation shown in Figure 8 between the ratio $E_{elst}/E_{disp}$ of the *L*-ArBeO, and the MEP of *L*. In any case, not unexpectedly, the *absolute* values of $E_{disp}$ were totally uncorrelated with respect to the MEP.

### 4.4. Bonding Analysi of the LAr and the L-ArBeO

In keeping with the results of the SAPT analysis, all the presently-investigated *L*Ar and *L*-ArBeO were assigned as nCov(C). Likewise, the exemplary N_2_-Ar and linear N_2_-ArBeO shown in Figure 1, the H^−^(**r**) regions of any L and Ar or ArBeO, were, in fact, invariably well separated and plunged in a zone of positive values of H(**r**); the latter wasalso invariably positive over Ω_s_. However, as shown in Table 3, when going from any *L*Ar to the corresponding *L*-ArBeO, the bond indices, particularly the N(Ω_s_) and the ρ_s_(ave), underwentappreciable changes that were, in particular, clearly related to the changes occurring in their SAPT terms (vide supra). Thus, for the Ng-ArBeO (Ng = He, Ne, Ar), the linear N_2_-ArBeO, OC-ArBeO, CO-ArBeO, ClF-ArBeO, and HF-ArBeO, the T-shaped F_2_-ArBeO and Cl_2_-ArBeO, and the C_3v_ isomer H_3_N-ArBeO, the values of both N(Ω_s_) and ρ_s_(ave) were higher than those predicted for the corresponding *L*Ar. This suggests tighter interactions, well consistent with the SAPT prediction of *L*-ArBeO complexes more stable than the *L*Ar. On the other hand, for the T-shaped N_2_-ArBeO, and the linear F_2_-ArBeO and Cl_2_-ArBeO, the values of the two bond indices werelower than those predicted for the corresponding *L*Ar, in good agreement with the lower stability of the *L*-ArBeO predicted by the SAPT analysis. This confirms the sensitivity of our proposed method in catching even subtle differences in the bonding character of the Ng complexes.

## 5. Conclusions 

The comparison of the *L*-ArBeO with their “dispersive” cousins *L*Ar allowed us to highlight structural and stability effects peculiarly related to the *σ*-hole of ArBeO. Consistent with the electrostatic model of NCIs [16,59], the most-affected SAPT term wasthe $E_{elst}$. The values predicted for the *L*-ArBeO were clearly related to the values of the MEP at the V_S,max_ or V_S,min_ of L, and ranged from negative (attractive) to positive (repulsive) values depending on the negative/positive value of the MEP. The percentage contribution of $E_{elst}$ may actually arrive up to ca 80% of the attractive interaction, and decisively determined the overall increased or decreased stability of any *L*-ArBeO with respect to the *L*Ar. The positive σ-hole of ArBeO also increased the inductive component of the interactionby an extent roughly related to the polarizability of L. The latter effects were, however, less pronounced than those occurring for the electrostatic component, the $E_{ind}$ accounting for only ca. 9−21% of the overall stabilization. The nearly constant values of $E_{ind}\left(\%\right)$ actually resulted in a regular inverse dependence between the electrostatic and the dispersive component, even though the absolute contribution of $E_{disp}$ wasnot directly correlated with the MEP of L. All these major effects become even clearer if one considers the bonding situation of the *L*Ar. In the absence of the σ-hole on Ar, the contribution of the electrostatic component wasonly minor, and nearly constant at around ca. 12−22% of the attractive stabilization. The interaction was, indeed, invariably dominated by the dispersion, even though the nearly null value of the electrostatic potential of Ar allowed us to recognize the direct influence of the MEP of L on the inductive component of the interaction. Our reached conclusions could be valid for other types of NCIs, especially those involving σ-holes of respectable strength such as those occurring in ArBeO. In this regard, there is, certainly, room for further future investigation.

## Figures and Tables

**Figure 1 molecules-26-04477-f001:**
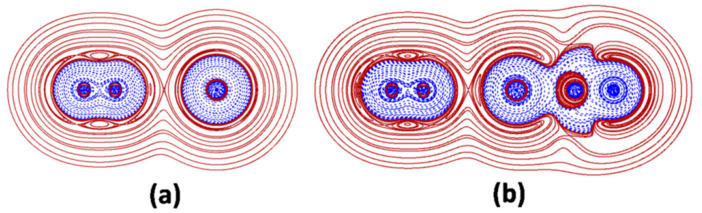
2D-plots of *H*(***r***) in the main plane of (**a**) N_2_-Ar and (**b**) linear N_2_-ArBeO (solid/brown and dashed/blue lines correspond, respectively, to positive and negative values).

**Figure 2 molecules-26-04477-f002:**
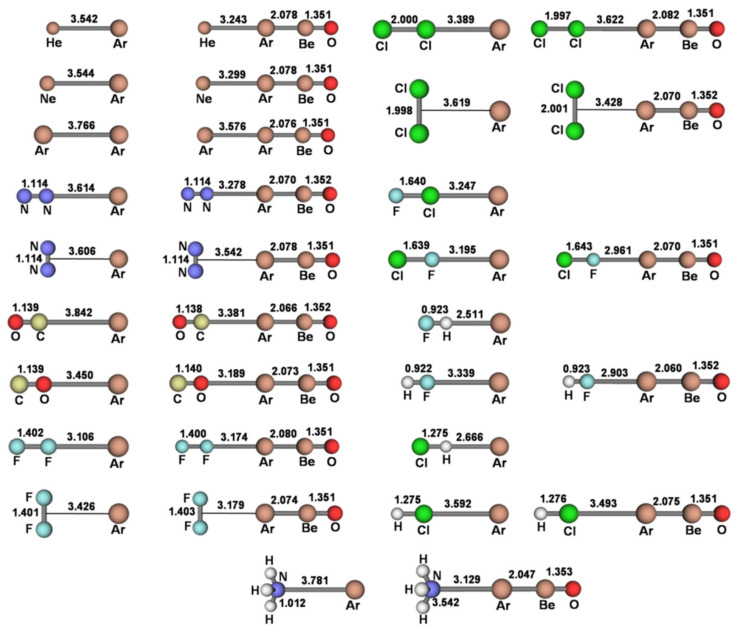
MP2/aVTZconnectivities and bond distances (Å) of the *L*Ar and *L*-ArBeO (*L* = He, Ne, Ar, N_2_, CO, F_2_, Cl_2_, ClF, HF, HCl, NH_3_).

**Figure 3 molecules-26-04477-f003:**
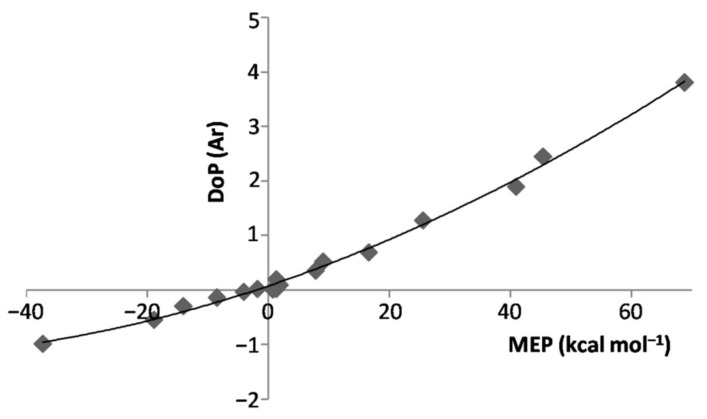
DoP(Ar) of the *L*Ar vs. MEP at the *V*_S,max_/*V*_S,min_ of *L*.

**Figure 4 molecules-26-04477-f004:**
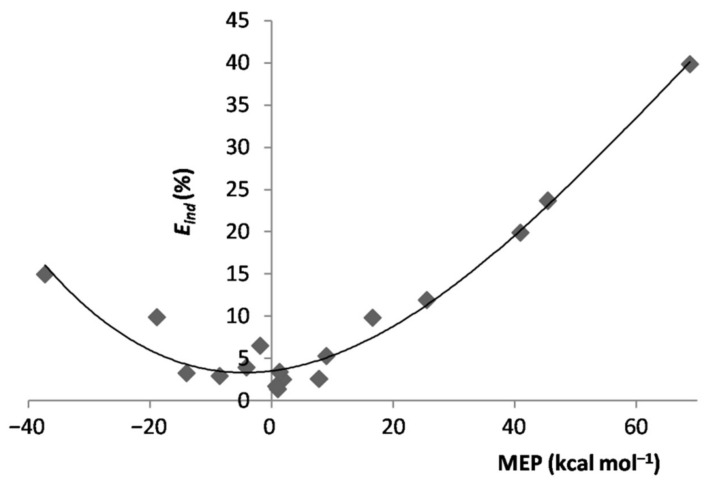
$E_{ind}\left(\%\right)$ of the LAr vs. MEP at the V_S,max_/V_S,min_ of L.

**Figure 5 molecules-26-04477-f005:**
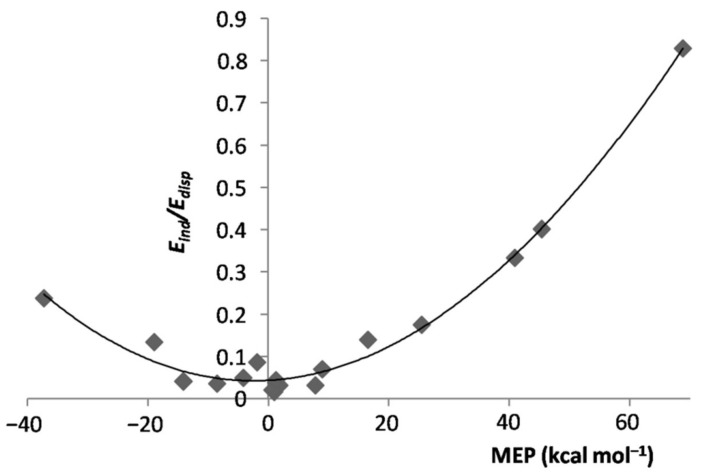
$E_{ind}/E_{disp}$ ratio of the LArvs. MEP at the V_S,max_/V_S,min_ of L.

**Figure 6 molecules-26-04477-f006:**
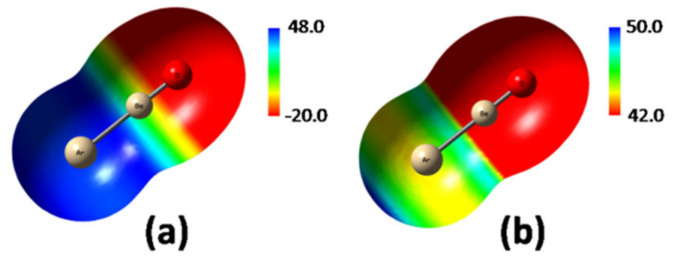
MP2/aVTZ MEP (0.0010 *ea*_0_^−3^ isodensity surface) of ArBeO plotted using a wider (**a**) and narrower (**b**) scale (values in kcal mol^−1^).

**Figure 7 molecules-26-04477-f007:**
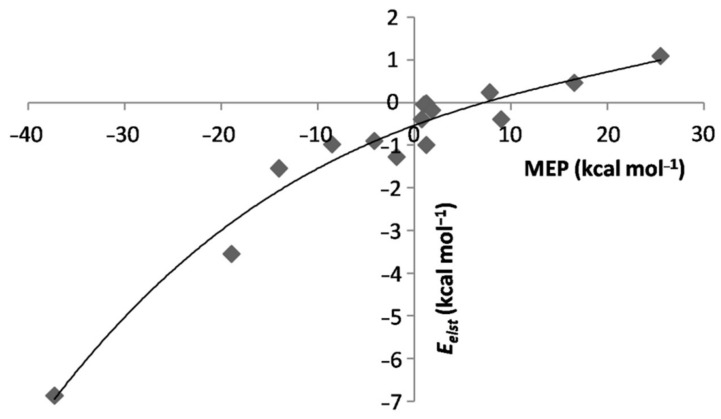
$E_{elst}$ of the *L*-ArBeO vs. MEP at the *V*_S,max_/*V*_S,min_ of *L*.

**Figure 8 molecules-26-04477-f008:**
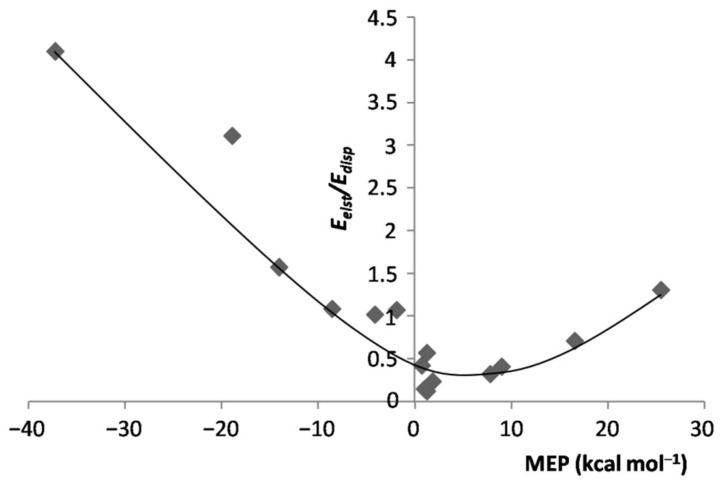
$E_{elst}/E_{disp}$ ratio of the *L*-ArBeOvs. MEP at the *V*_S,max_/*V*_S,min_ of *L*.

**Table 1 molecules-26-04477-t001:** SAPT analysis (kcal mol^−1^) of the *L*Ar(dissociation limits: *L* + Ar) and *L*-ArBeO(dissociation limits: *L* + ArBeO) (see Figure 2) performed with the aVTZ/mbf basis set. *E_elst_*(%), *E_ind_*(%), and *E_disp_*(%) are the percentage contributions, respectively, of |*E_elst_*|, |*E_ind_*|, and |*E_disp_*| to (|*E_elst_*|+ |*E_ind_*| + |*E_disp_*|).

	$E_{elst}$	$E_{ind}$	$E_{disp}$	$E_{exch}$	$E_{int}^{SAPT}a$	Lit.	$E_{elst}(\%)$	$E_{ind}(\%)$	$E_{disp}(\%)$	DoP(Ar)b
**HeAr**	−0.0119	−0.0040	−0.1049	0.0638	−0.0570	−0.0588 ^c^	9.85	3.35	86.80	0.026
**He-ArBeO**	−0.0204	−0.0432	−0.1686	0.1218	−0.1104		8.80	18.61	72.59	
**NeAr**	−0.0433	−0.0037	−0.2306	0.1534	−0.1242	−0.1342 ^d^	15.61	1.34	83.05	0.039
**Ne-ArBeO**	−0.0507	−0.0797	−0.3371	0.2561	−0.2114		10.85	17.04	72.11	
**ArAr**	−0.1332	−0.0179	−0.5641	0.4311	−0.2841	−0.2846 ^e^	18.62	2.50	78.88	0.099
**Ar-ArBeO**	−0.1745	−0.2428	−0.7345	0.6422	−0.5096		15.15	21.08	63.77	
**N_2_−Ar**	−0.1074	−0.0189	−0.5284	0.4039	−0.2508	−0.2221 ^f^	16.41	2.88	80.71	−0.14
**N_2_-ArBeO**	−0.9782	−0.3132	−0.9032	1.0386	−1.1560		44.57	14.27	41.16	
**N_2_-Ar (T** ^g^ **)**	−0.1635	−0.0223	−0.6912	0.5682	−0.3088	−0.2913 ^f^	18.64	2.54	78.82	0.36
**N_2_-ArBeO (T)**	0.2296	−0.1944	−0.7061	0.5104	−0.1605		20.32	17.20	62.48	
**OC-Ar**	−0.1063	−0.0191	−0.4698	0.3506	−0.2446	−0.2078 ^h^	17.86	3.21	78.93	−0.29
**OC-ArBeO**	−1.5489	−0.3531	−0.9869	1.3982	−1.4907		53.61	12.22	34.16	
**CO-Ar**	−0.1224	−0.0290	−0.5916	0.4618	−0.2812	−0.2391 ^h^	16.47	3.91	79.62	−0.034
**CO-ArBeO**	−0.9018	−0.3204	−0.8840	0.8791	−1.2271		42.81	15.21	41.97	
**F_2_-Ar**	−0.2343	−0.1155	−0.8264	0.8015	−0.3747	−0.3510 ^i^	19.92	9.82	70.26	0.69
**F_2_-ArBeO**	0.4667	−0.2444	−0.6561	0.3806	−0.0532		34.14	17.86	48.00	
**F_2_-Ar (T)**	−0.1406	−0.0133	−0.6521	0.4774	−0.3286	−0.3146 ^i^	17.44	1.65	80.91	0.013
**F_2_-ArBeO (T)**	−0.3956	−0.2249	−0.9411	0.8385	−0.7231		25.33	14.40	60.27	
**Cl_2_-Ar**	−0.4113	−0.2373	−1.3498	1.3329	−0.6655	−0.6487 ^j^	20.58	11.87	67.54	1.28
**Cl_2_-ArBeO**	1.0857	−0.3846	−0.8325	0.3932	0.2618		47.15	16.70	36.15	
**Cl_2_-Ar (T)**	−0.3778	−0.0603	−1.3592	1.1461	−0.6512	−0.6314 ^j^	21.02	3.35	75.63	0.20
**Cl_2_-ArBeO (T)**	−0.9942	−0.6121	−1.7521	1.7508	−1.6076		29.60	18.22	52.17	
**FCl-Ar**	−0.5476	−0.5337	−1.6006	1.8462	−0.8357	−0.8101 ^k^	20.42	19.90	59.68	1.89
**ClF-Ar**	−0.2064	−0.0716	−0.8296	0.6875	−0.4201	−0.3687 ^k^	18.63	6.47	74.90	0.021
**ClF-ArBeO**	−1.2759	−0.4951	−1.1938	1.2087	−1.7561		43.04	16.70	40.27	
**FH-Ar**	−0.2437	−0.8019	−0.9663	1.4098	−0.6020	−0.6037 ^l^	12.11	39.86	48.03	3.80
**HF-Ar**	−0.1203	−0.0705	−0.5234	0.4182	−0.2960	−0.3067 ^l^	16.84	9.87	73.29	−0.54
**HF-ArBeO**	−3.5413	−0.4587	−1.1374	1.6639	−3.4735		68.93	8.92	22.14	
**ClH-Ar**	−0.3268	−0.4413	−1.0998	1.3561	−0.5118	−0.5050 ^m^	17.50	23.63	58.87	2.45
**HCl-Ar**	−0.2153	−0.0611	−0.8788	0.7188	−0.4364	−0.4288 ^m^	18.64	5.29	76.07	0.52
**HCl-ArBeO**	−0.3947	−0.3613	−0.9692	0.7563	−0.9689		22.88	20.94	56.18	
**H_3_N-Ar**	−0.2052	−0.1364	−0.5726	0.5809	−0.3333	−0.2966 ^n^	22.45	14.92	62.63	−0.98
**H_3_N-ArBeO**	−6.8566	−1.0230	−1.6735	4.1151	−5.4380		71.77	10.71	17.52	

^a^
$E_{int}^{SAPT}=E_{elst}+E_{ind}+E_{disp}+E_{exch}$
; ^b^ defined by Equation (4); ^c^ Ref. [45]; ^d^ Ref. [46]; ^e^ Ref. [47]; ^f^ Ref. [48]; ^g^ T-shaped; ^h^ Ref. [49]; ^i^ Ref. [50]; ^j^ Ref. [51]; ^k^ Ref. [52]; ^l^ Ref. [53]; ^m^ Ref. [54]; ^n^ Ref. [55].

**Table 2 molecules-26-04477-t002:** MP2/aVTZ properties of the nCov(C) bonds (see text) of the *L*Ar and *L*-ArBeO (see Figure 2). *N*(*Ω_s_*), *ρ*_s_(ave), and *H_s_*(ave/max/min) are, respectively, the total electronic charge (m*e*), the average electron density (*e a*_0_^−3^), and the average, maximum, and minimum of *H*(***r***) (hartree *a*_0_^−3^) over the volume *Ω_s_* (*a*_0_^3^) enclosed by the *s*(***r***) = 0.4 isosurfaceat around the BCP.

Bond		*Ω_s_*	*N*(*Ω_s_*)	*ρ*_s_(ave)	*H_s_*(ave/max/min)
**He-Ar**	HeAr	0.0212	0.024	0.0011	0.00040/0.00042/0.00038
	He-ArBeO	0.0234	0.043	0.0018	0.00079/0.00082/0.00076
**Ne-Ar**	NeAr	0.0316	0.063	0.0020	0.00059/0.00061/0.00057
	Ne-ArBeO	0.0372	0.11	0.0030	0.00084/0.00103/0.00090
**Ar-Ar**	ArAr	0.0948	0.27	0.0029	0.00074/0.00078/0.00069
	Ar-ArBeO	0.0942	0.35	0.0038	0.00113/0.00122/0.00105
**N-Ar**	N_2_-Ar	0.0788	0.22	0.0028	0.00076/0.00080/0.00071
	N_2_-ArBeO	0.0904	0.46	0.0051	0.00162/0.00172/0.00150
	N_2_-Ar (T ^a^)	0.1570	0.46	0.0029	0.00075/0.00082/0.00067
	N_2_-ArBeO (T)	0.1344	0.40	0.0030	0.00086/0.00093/0.00078
**C-Ar**	OC-Ar	0.0914	0.21	0.0023	0.00060/0.00063/0.00056
	OC-ArBeO	0.1203	0.64	0.0053	0.00150/0.00167/0.00139
**O-Ar**	CO-Ar	0.0736	0.24	0.0032	0.00086/0.00091/0.00080
	CO-ArBeO	0.0785	0.40	0.0051	0.00159/0.00168/0.00148
**F-Ar**	F_2_-Ar	0.0966	0.46	0.0047	0.00154/0.00176/0.00137
	F_2_-ArBeO	0.0686	0.24	0.0035	0.00124/0.00134/0.00113
	F_2_-Ar (T)	0.2052	0.59	0.0029	0.00068/0.00076/0.00058
	F_2_-ArBeO (T)	0.2254	0.95	0.0042	0.00111/0.00126/0.00096
**Cl-Ar**	Cl_2_-Ar	0.1687	0.90	0.0053	0.00153/0.00165/0.00139
	Cl_2_-ArBeO	0.1139	0.32	0.0028	0.00095/0.00101/0.00087
	Cl_2_-Ar (T)	0.4444	1.56	0.0035	0.00088/0.00095/0.00076
	Cl_2_-ArBeO (T)	0.4482	2.08	0.0046	0.00127/0.00142/0.00111
**Cl-Ar**	FCl-Ar	0.2167	1.44	0.0067	0.00184/0.00199/0.00165
**F-Ar**	ClF-Ar	0.0866	0.38	0.0043	0.00121/0.00132/0.00111
	ClF-ArBeO	0.0920	0.61	0.0066	0.00212/0.00233/0.00193
**H-Ar**	FH-Ar	0.0777	0.65	0.0084	0.00148/0.00158/0.00134
**F-Ar**	HF-Ar	0.0700	0.23	0.0032	0.00092/0.00097/0.00086
	HF-ArBeO	0.0936	0.73	0.0078	0.00259/0.00286/0.00234
**H-Ar**	ClH-Ar	0.0922	0.64	0.0069	0.00129/0.00140/0.00123
**Cl-Ar**	HCl-Ar	0.1268	0.46	0.0037	0.00112/0.00120/0.00102
	HCl-ArBeO	0.1170	0.47	0.0041	0.00135/0.00145/0.00124
**N-Ar**	H_3_N-Ar	0.1113	0.31	0.0028	0.00055/0.00063/0.00051
	H_3_N-ArBeO	0.1815	1.65	0.0091	0.00185/0.00209/0.00168

^a^ T-shaped.

**Table 3 molecules-26-04477-t003:** MP2/aVTZ values (kcal mol^−1^, 0.0010 *ea*_0_^−3^ isodensity surface) at the minimum (*V_S,Min_*) or maximum (*V_S,Max_*) points of the MEP calculated at the MP2/aVTZ optimized geometries (Å and °), and experimental polarizabilities α (Å^3^) of *L*.

L	MEP point		R/θ	$\bar{\alpha }$	${\alpha_{∥}}^{a}$	${\alpha_{⊥}}^{b}$
**N_2_**	V_S,Min_(N): −8.53	V_S,Max_(perp): 7.82	1.114	1.75 ^c^	2.20 ^d^	1.52 ^d^
**CO**	V_S,Min_(C): −14.04	V_S,Min_(O): −4.12	1.139	1.95 ^e^	2.31 ^e^	1.77 ^e^
**F** _2_	V_S,Max_(F): 16.57	V_S,Max_(perp): 0.76	1.401	1.25 ^f^	1.84 ^f^	0.96 ^f^
**Cl_2_**	V_S,Max_(Cl): 25.50	V_S,Max_(perp): 1.26	1.999	4.59 ^g^	6.27 ^g^	3.75 ^g^
**ClF**	V_S,Max_(Cl): 40.94	V_S,Max_(F): −1.86	1.639	2.68 ^h^	3.37 ^i^	2.34 ^j^
**HF**	V_S,Max_(H): 68.78	V_S,Max_(F): −18.91	0.922	0.83 ^k^	0.94 ^k^	0.77 ^k^
**HCl**	V_S,Max_(H): 45.38	V_S,Max_(Cl): 9.00	1.275	2.58 ^l^	2.74 ^l^	2.50 ^l^
NH_3_	V_S,Min_(N): −37.25		1.012/106.8	2.15 ^m^		

^a^ Component along the bond axis; ^b^ component perpendicular to the bond axis; ^c^ calculated as $\bar{\alpha }=\frac{\left(\alpha_{∥}+\;2\;\alpha_{⊥}\right)}{3}$; ^d^ Ref. [58];^e^ Ref. [59]; ^f^ Ref. [60]; ^g^ Ref. [61]; ^h^ evaluated by the empirical method proposed in Ref. [62]; ^i^ Ref. [63]; ^j^ calculated as $\alpha_{⊥}=\frac{\left(3\bar{\alpha }-\alpha_{∥}\right)}{2}$; ^k^ Ref. [64]; ^l^ Ref. [65]; ^m^ Ref. [66].

## Data Availability

Not applicable.

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
