# Peer review of "Concerning the Role of σ-Hole in Non-Covalent Interactions: Insights from the Study of the Complexes of ArBeO with Simple Ligands"

_molecules, 2021, doi:10.3390/molecules26154477_

Round 1

Reviewer 1 Report

read pdf

Reviewer 2 Report

This manuscript presents theoretical study of the σ-hole effect of Ar atom in ArBeO based on a series of interactions between LAr and L-ArBeO. Considering that results are reliable and interesting and the topic if of high importance, this manuscript can be recommended for publication in Molecules after minor corrections. 

1) In the second part of “Methods of Bonding Analysis”, if the description of three types of interactions A, B or C can be illustrated by figures, it will be more convenient for readers to understand. Otherwise, the text description is a little bit too technical and not easy to understand for non-professionals. The same is true for the Abstract.

2) In 4.3 “From the LAr to the L-ArBeO: the role of the σ-hole of ArBeO”, the authors mentioned the MPE in Figure 5. In my opinion, if MEP figures can be modified into a translucent ones in which specific molecules can be seen, it will help readers to more clearly see what atoms are corresponding to the positive and negative regions.

Round 2

Reviewer 1 Report

The revised version incorporates my suggestions and provides new insights for readers.